# Climate Change Knowledge and Perception among Farming Households in Nigeria

**Mustapha Yakubu Madaki** [1,2] 🆔**, Steffen Muench** [1] 🆔**, Harald Kaechele** [2,3] **and Miroslava Bavorova** [1,*]

1 Department of Economics and Development, Faculty of Tropical AgriSciences,
  Czech University of Life Sciences Prague, 16500 Prague, Czech Republic;
  yakubu_madaki@ftz.czu.cz (M.Y.M.); muench@ftz.czu.cz (S.M.)
2 Leibniz Centre for Agricultural Landscape Research (ZALF), Eberswalder Straße 84,
  15374 Müncheberg, Germany; harald.kaechele@zalf.de
3 Department of Environmental Economics, Eberswalde University for Sustainable Development,
  Schicklerstraße 5, 16225 Eberswalde, Germany
* Correspondence: bavorova@ftz.czu.cz

**Abstract:** Nigeria is committed to achieving a 20% unconditional and 45% conditional reduction of GHG emissions by 2030 through a strong focus on awareness of and preparedness for climate change impacts via the mobilization of local communities for climate change mitigation actions. As land cover changes and forestry contribute 38% and agriculture contributes 13% of the country's GHGs, farmers are among the stakeholders to be aware of and prepare for climate change mitigations and adaptations. This study assessed the knowledge of agriculturally related practices associated with climate change and its relation to climate change perception. One thousand and eighty (1080) smallholder farmers were interviewed across six agroecological zones (AEZs) of Nigeria using a structured questionnaire. The results revealed that most farmers know that deforestation and land clearance by bush burning contributes to climate change. However, many farmers did not know that methane emissions from livestock (enteric fermentation) can cause climate change. Our results further show that the farmers' perception of climate change is associated with climate change knowledge. Factors affecting the climate change knowledge of farmers include information received from government extension services, environmental NGOs, and radio, as well as experiencing extreme weather events. Farmers of dry AEZs were more aware and knowledgeable of the agricultural practices contributing to the changing environment. Increased exposure to climate change events thus appears to elevate the knowledge on the topic. Using government services, environmental NGOs, and radio to disseminate climate change information will help further in guiding and shaping farmers' perceptions towards scientific findings for appropriate actions.

**Keywords:** causes of climate change; climate change perception and knowledge; climate change mitigation; farm practices; knowledge gap theory; agroecological zones; Nigeria

## 1. Introduction

Nigeria is identified as a climate change hotspot, with an expected increase in unpredictable rainfall patterns, flooding, and drought [1]. Being Africa's largest economy, Nigeria is committed to reducing GHG emissions to minimize climate change impacts on a regional scale [2]. In addition, climate change jeopardizes the economic well-being of Nigeria's population, which is reflected in projected GDP losses of 1.27% by 2027 and 3.42% by 2037 [3]. As a condition of the Paris Agreement, Nigeria formulated an Intended Nationally Determined Contribution (INDC) to the United Nations Framework Convention on Climate Change (UNFCCC) to achieve a 20% unconditional and 45% conditional reduction of GHG emissions by 2030 [4]. Furthermore, the INDC focuses on awareness of climate change impacts via the mobilization of local communities for climate change adaptation action [5].

There is considerable variation in the awareness and risk perception among societies in different parts of the world [6] due to cultural factors, including worldviews and political ideology [7]. Climate change risk perceptions and mitigation behaviors are shaped by a wide array of factors, such as socio-demographic characteristics [8], psychological factors, and mental models [9]. In addition, the climate change risk perception model (CCRPM) discussed by van der Linden [10] includes personal experience with extreme weather events as a reliable predictor in assessing an individual's risk perception towards climate change.

Despite most people being aware of climate change, a deeper understanding of their experiential mindset is necessary to reveal their attitude toward climate change mitigation measures [11]. As climate change effects can differ even within one country, it is of utmost importance to research this issue locally. Global food systems are responsible for up to 37% of the total GHG emissions in the world [12]. Approximately 24% of the total global GHG emissions were caused by the agricultural sector in 2010 alone [13].

In Nigeria, land cover changes and forestry contribute 38.2%, while agriculture contributes 13% of national GHG emissions, with an increase of 25% between 1990 and 2014 [14]. Investigating the awareness and knowledge of climate change causes among farmers in the context of conceptualizing appropriate mitigation efforts is thus of the greatest relevance. However, so far, few studies have elicited such information [15]. Existing studies that investigate climate change perception among farmers in various parts of the world undermine the paramount importance of such findings in the policy-making process [16,17].

While the awareness level of the causes and effects of climate change among farmers in Nigeria has been studied [18], the relationship between the perceived climate change impact of a farmer and a farmer's knowledge of agricultural climate change causes has not yet been analyzed. The various agroecological zones (AEZs) of Nigeria, where agricultural dependence is high and climatic conditions significantly vary, are deemed to be an ideal research area for investigating how the knowledge of climate change is associated with farmers' climate change perceptions. Shedding light on this issue serves as the basis to guide and shape farmers' perceptions for more tailored climate change mitigation and adaptation policies. This research gap served as motivation for our study.

Within our research, climate change awareness of farmers indicates that the farmer was aware of the term "climate change" [15,19]. Farmers' climate change perception has three dimensions: First, how farmers understand the term "climate change", second, how farmers conceptualize what constitutes climate change, and third, how the experience of climate change shapes their attitude toward the concept [15]. Experience demonstrates that small-scale farmers are not concerned with questions related to causes and effects but rely more on their own perception and awareness of changes [20]. Farmers respond to climate change according to their perception of the causes of environmental changes rather than scientific facts and evidence [21,22].

As pointed out earlier, the awareness of climate change among farmers has been a focus of interest in recent studies in various parts of the world [18,23–28]. Farmers' climate change awareness and climate risk perception has be reported to influence their climate change adaptation behaviors [29–31]. According to the knowledge gap theory, segments of the population with higher socioeconomic status tend to acquire information about a subject at a faster rate than lower-status segments. The gap in knowledge between these segments thus tends to increase rather than decrease [32]. In this way, farmers with high social status are likely to be more knowledgeable of climate change, as they have better access to a variety of information sources/channels that broadcast or publish governmental and non-governmental programs on climate change. This indicates the effect of socio-economic variables such as education, income, etc., as well as the role of information sources and channels in the knowledge of climate change of farmers.

However, some authors found that people with low socioeconomic status are more knowledgeable about local issues that affect them directly than their counterparts [15,33]. This particularly applies to small-scale farmers with poor coping strategies and financial shock absorbers, and this depicts the effect of climate risk experience in climate risk-prone

AEZs such as dry agroecology (arid, semi-arid, savannah zones, etc.). Based on these findings, our study analyzed the climate change knowledge of farmers and its association with their perception and sought to provide answers to the following research questions: (i) Is the climate change knowledge of farmers associated with their climate change perception? (ii) Which factors affect the awareness of climate change and the farmers' knowledge of the causes of climate change?

## 2. Conceptual Framework

We conceptualized the factors that can explain the climate change knowledge of farmers based on the knowledge gap theory by grouping our independent variables into four categories: (i) socioeconomic characteristics, (ii) climate change information sources, (iii) climate change information channels, and (iv) climate change experience of farmers (Figure 1).

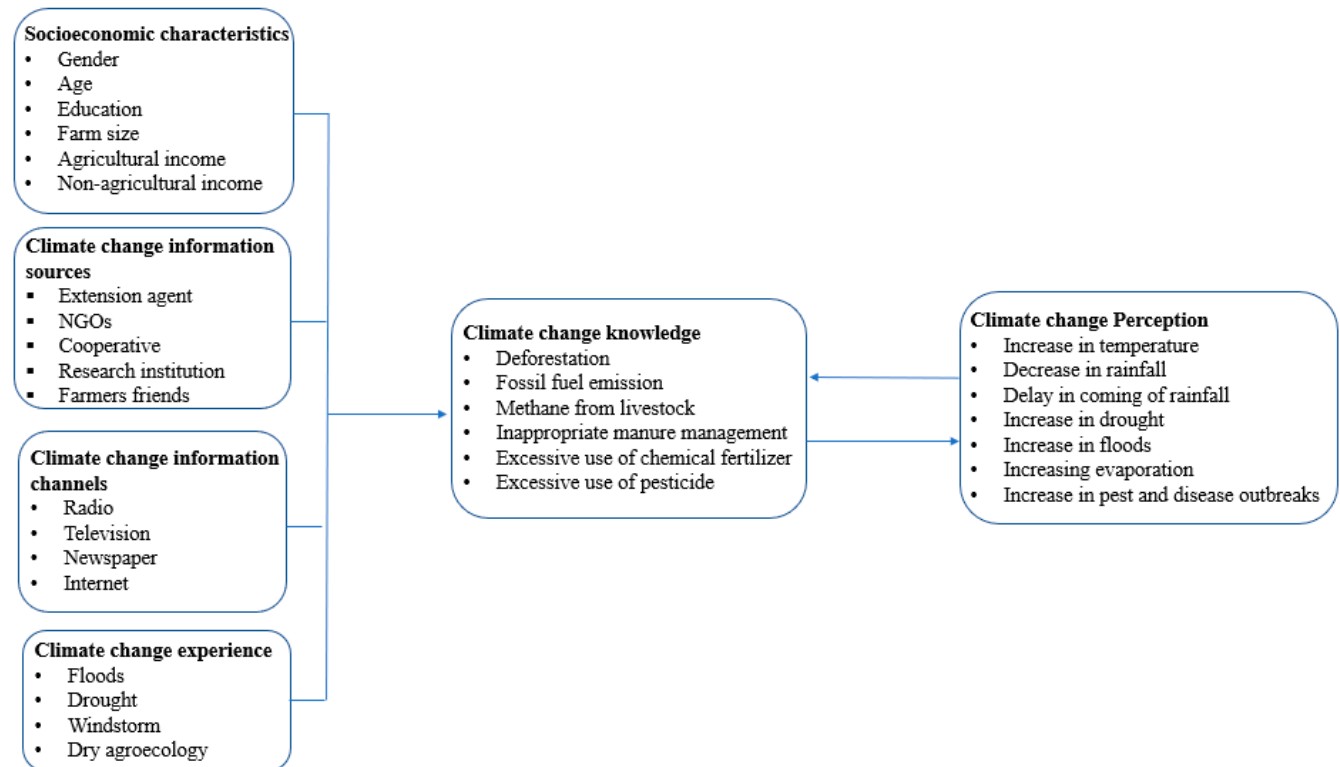

**Figure 1.** Conceptual framework of climate change knowledge of farmers.

## 3. Methodology

### 3.1. Study Area

Nigeria has a total land area of 910,768 km$^2$ and a water area of 13,000 km$^2$ [34]. The country is characterized by a tropical climate, with six distinctive AEZs. These AEZs can be categorized into the Semi-arid zone, the Sudan savanna, the Guinea savanna, the Swamp forest, the Mangroves, and the Rainforest. Rainfall is bimodal in the humid/southern part (freshwater swamp, Mangroves, and Rainforest) and unimodal in the dry/northern part (the Semi-arid zone and the Guinea and Sudan savannas) of the country [35]. Annual rainfall varies significantly from about 500 mm/year in the north (the Semi-arid zone) to 3000 mm/year in the extreme south (the Mangrove and Rainforest ecological zones). The humid climate is a result of the proximity to the Gulf of Guinea. Seasonal temperature differences range from 40 °C in the extreme north (the Semi-arid zone) around April and May to only 12 °C in the central part of the country (the drive savannah agroecological zones) around December and January [35]. The mean temperature of the country has continually increased over the last 30 years, and the mean precipitation of the country

has decreased [36]. The drought occurrences are more pronounced in the dry AEZs [37], and floods affect almost all parts of the country, and are especially prevalent in the dry AEZs [38].

### 3.2. Sampling Procedure and Sample Size

A multi-stage sampling procedure was used to select the respondents for this study, in order to have a representative sample. In the first stage, we applied a convenience sampling of one state from each AEZ because of the insecurity challenges in some states (Figure 2), followed by the random sampling method (a lottery) in the second stage, which was used to select a total of 12 local government areas for an equal chance of participation. Based on these specifications, in the third stage, two wards were selected randomly from each local government area, resulting in a total number of 24 wards. Lastly, in the fourth stage, 45 farming households were drawn randomly (again using a lottery) to have a representative sample for the study population from each selected ward, reaching a total of 1080 farming households for the study. In cases where random sampling was not possible because of missing lists of farmers (about 20% of wards), snowball sampling was used.

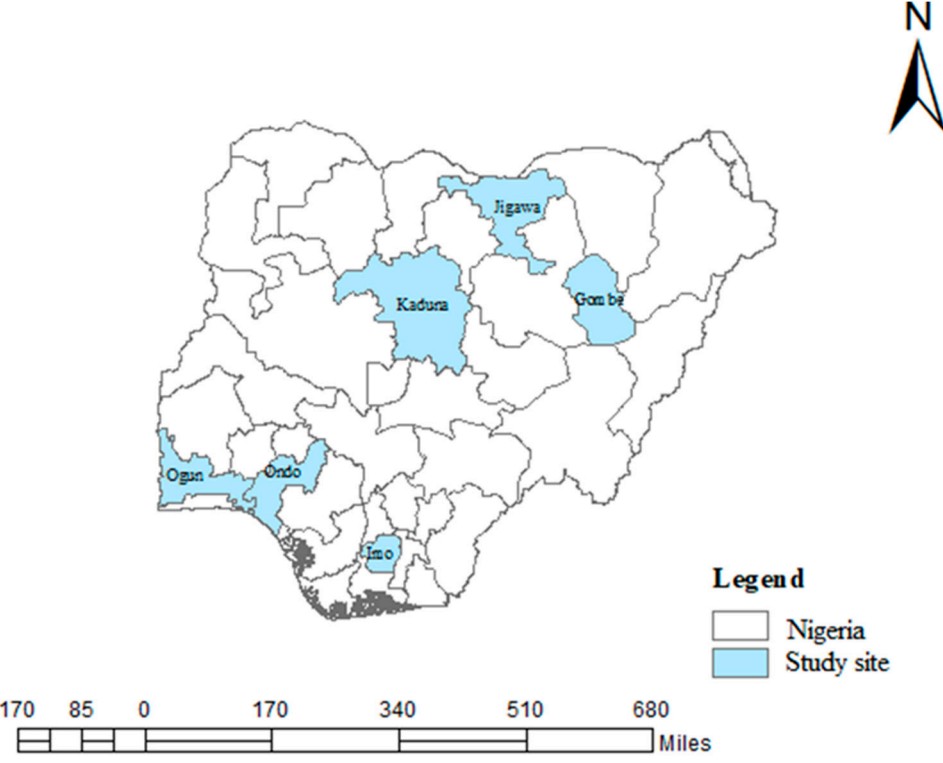

**Figure 2.** Map of Nigeria showing the study sites.

### 3.3. Data Collection

Primary data were collected with the help of 12 trained enumerators using a questionnaire/pen and paper survey between October 2020 and February 2021. Household heads or their representatives (less than 10% of respondents) were interviewed. Most of the interviews conducted in native languages (Hausa, Yoruba, and Igbo), and responses were translated into English on the spot. A pre-test survey was conducted with 40 farmers, and modifications were made based on the pre-test outcome prior to data collection. Most questions were derived from the knowledge gap theory as used in the literature [25,27,39,40] and adjusted to suit regional differences accordingly. The questions included respondents' weather information sources (e.g., government agencies, NGOs, research institutions, other farmers), information channels (e.g., radio, television, newspapers, internet), climate risk event experiences (e.g., drought, floods) and socioeconomic characteristics (e.g., household,

farm, and institutional characteristics), and climate change awareness/knowledge of causes and indicators of climate change, such as increases in temperature and evaporation as well as rainfall variability. Based on meteorological data homogeneity, the data were grouped into dry cluster AEZs (Semi-arid, Sudan savannah, and Guinea savannah) and humid cluster AEZs (Tropical rainforest, Mangrove, and Swamp forest).

*3.4. Data Analysis*

3.4.1. Correlation Analysis

Spearman correlation was used to assess the relationship between climate change knowledge and the climate change perception of farmers. The climate change knowledge score (ranging between 0 and 7) was correlated with the perception of climate change indicators in the study area, such as an increase in temperature, decline in the amount of annual rainfall, delay in rainfall start, etc. (measured on a five-point scale). The Spearman correlation equation is:

$$\rho = 1 - \frac{6 \sum d^2}{n(n^2 - 1)} \tag{1}$$

where $\rho$ is the Spearman rank correlation coefficient, $d^2$ is the difference between the rank value of the climate change knowledge score and the climate perception of farmers, and $n$ is the number of observations.

3.4.2. Probit Model: Climate Change Awareness

To examine the factors influencing climate change awareness, a binary response (Probit) model was used. Following previous studies, we considered that a farming household head was aware of climate change if they heard the term climate change from information sources and channels or if the farmer experienced climate risk events, such as frequent drought and floods, that impacted farm productivity [26–28]. The mathematical expression of the relationship of logit regression in its non-linear form (sigmoid curve) is:

$$\frac{P_i}{1 - P_i} = \frac{1 + \exp(Z_i)}{1 + \exp(-Z_i)} \tag{2}$$

$$L_i = ln\left[\frac{Z_i}{P(1 - P_i)}\right] Z_i \tag{3}$$

In Equation (3), $P_i/(1 - P_i)$ is the probability ratio that the farmer will be aware of climate change. If a farmer is aware, then the value of the dependent variable is 1; it is 0 otherwise.

When it transforms into a linear appearance:

$$Ln(y_{i1}) = \alpha + \beta_1 X_1 + \beta_2 \beta_2 + \beta_3 X_3 \ldots + \beta_{20} X_{20} + \varepsilon \tag{4}$$

In Equation (4), $y_{i1}$ is the probability that farming household head $i$ will be aware of climate change by obtaining climate information or experiencing climate variability being greater than zero ($y_{i1} > 0$, aware = 1, otherwise = 0), $\alpha$ is a constant, $\beta_1 - \beta_{20}$ is the regression coefficient, $X_i - X_{20}$ denotes the set of explanatory variables or factors that influence climate change awareness, $\varepsilon$ is the error term.

Factors that affect climate change awareness were identified according to the reviewed literature, as shown in Table 1. Gender, farming experience, and information usage showed varying effects in previous research, indicating the importance of considering regional differences within this context [23,41]. Education and farming experience were found to influence the climate change awareness of farmers [23,26,41]. However, government extension services show mixed effects on the degree of climate change awareness among farmers [23,26,28,42,43]. The most reliable sources of farmers' climate change awareness were government extension agents, radio, the internet, magazines, newspapers, and television [44–47].

**Table 1.** Description of variables imported into the models (N = 1080).

| Variable | Description | Mean and Standard Deviation |
|---|---|---|
| *Dependent variables* | | |
| Climate change awareness | Yes = 1, otherwise = 0 | 0.72 (0.44) |
| Knowledge of climate change causes | Farmer's quiz score 0–7 | 2.62 (1.56) |
| *Independent variables* | | |
| *Sociodemographic characteristics* | | |
| Gender | Male = 1, female = 0 | 0.78 (0.41) |
| Age | Years | 48.15 (13.30) |
| Years of education | Years of formal education | 8.24 (5.59) |
| Farming experience | Years of being in farming | 22.61 (12.18) |
| Farm size | In hectare | 3.44 (3.45) |
| Agricultural income | Annual agricultural income (Naira) | 7563.60 (5249.34) |
| Non-agricultural income | Annual non-agricultural income (Naira) | 86.99 (96.78) |
| *Climate change information sources* | | |
| Government extension agent (GEA) | Receiving weather information from GEA (Yes = 1, No = 0) | 0.69 (0.45) |
| Environmental NGOs | Receiving weather information from NGOs (Yes = 1, No = 0) | 0.22 (0.42) |
| Farmers' cooperatives | Receiving weather information from farmers' cooperatives (Yes = 1, No = 0) | 0.37 (0.48) |
| University and research institution (URI) | Receiving weather information from URI (Yes = 1, No = 0) | 0.10 (0.31) |
| Farmers' friends | Receiving weather information from farmers' friends (Yes = 1, No = 0) | 0.40 (0.49) |
| *Climate change information channels* | | |
| Radio | Number of times receiving climate-related information via radio in a month | 9.84 (9.37) |
| Television | Number of times receiving climate-related information via television in a month | 1.63 (4.75) |
| Newspaper | Number of times receiving climate-related information via newspapers in a month | 0.49 (2.37) |
| Internet | Number of times receiving climate-related information via the internet in a month | 1.10 (4.46) |
| *Climate change experience* | | |
| Flooding | Number of flood experiences of farmer in the last 10 years | 0.73 (0.43) |
| Drought | Number of drought experiences of farmer in the last 10 years | 2.15 (2.23) |
| Windstorm | Number of windstorm experiences of farmer in the last 10 years | 0.71 (0.45) |
| Dry agroecological zones | If a farmer is from one of the three dry zones = 1, otherwise = 0 | 0.5 (0.50) |

### 3.4.3. Poisson Regression: Climate Change Knowledge

Poisson regression was used to analyze the factors affecting knowledge of agricultural practices contributing to climate change. Farmers were asked seven quiz questions on farming practices related to climate change mitigation to indicate their level of climate change knowledge. Each question answered correctly by a farmer received a score of 1, and a wrong answer or "I do not know" received 0. The scores for each farmer were summed up, with the final count score ranging from 0 to 7. Table 2 shows the score distribution of farmers in the seven areas that constituted the quiz questions:

**Table 2.** Farmers' scores on quiz questions of causes of climate change (N = 1080).

| Quiz Mark | Score Distribution of Farmers (%) | Cumulative Frequency |
|---|---|---|
| 0 | 10.11 | 10.11 |
| 1 | 9.46 | 19.57 |
| 2 | 29.13 | 48.70 |
| 3 | 25.88 | 74.58 |

**Table 2.** *Cont.*

| Quiz Mark | Score Distribution of Farmers (%) | Cumulative Frequency |
|---|---|---|
| 4 | 14.01 | 88.59 |
| 5 | 6.40 | 94.99 |
| 6 | 2.88 | 97.87 |
| 7 | 2.13 | 100.00 |

The seven quiz questions were based on the following topics.

i. Deforestation: this is the process of cutting down plants and crops. This breaks the carbon cycle by stopping the $CO_2$ absorption function of plants. Between 2015 and 2017, the global loss of tropical forests contributed to about 4.8 billion tonnes of $CO_2$ per year (or about 8–10% of annual human emissions of carbon dioxide) [48].

ii. Land clearance by bush burning: this is a process where farmers clear their farmlands using fire to prepare for the rainy season. Bush burning can deplete top-soil nutrients, potentially causing crop yields to decrease [49]. Furthermore, this practice changes organic nitrogen into mobile nitrates, which makes it very volatile and causes air pollution through the release of carbon stored in plant leaves, stems, and branches into the atmosphere [50].

iii. Fossil fuel use: this is the primary source of $CO_2$ emitted directly from human-induced impacts. The total $CO_2$ contribution from fossil fuel use and other industrial processes alone contributes 65% of global greenhouse gas emissions [13].

iv. Methane ($CH_4$) from livestock production: methane makes up the majority of emissions that come from farmed livestock, such as sheep and cattle. Animals naturally produce methane as a by-product of their digestive processes and release it into the air [51]. Between 1970 and 2010, emissions of $CH_4$ from enteric fermentation and rice cultivation increased by 20% [1].

v. Use of manure: inappropriate manure handling and application lead to the emission of $CH_4$ and Nitrous Oxide ($N_2O$); this agricultural activity contributes to climate change [13].

vi and vii. Use of chemical fertilizers and other agrochemicals: agricultural activities contribute approximately 30% of total greenhouse gas emissions, mainly due to the intensive use of chemical fertilizers and other agrochemicals [52].

The Poisson regression model is

$$\Pr(Y = y) = \frac{e^{-\pi v} \pi v^y}{y^i}$$

In this model, we typically assumed $Ln[\sum(y)] = Ln(\pi) + (v)$
Which is

$$y_{i2} = \alpha + \beta_1 X_1 + \beta_2 \beta_2 + \beta_3 X_3 \ldots + \beta_{20} X_{20} + \varepsilon \tag{5}$$

In Equation (5), $y_{i2}$ is the number of questions a farmer answered correctly (Table 2) with answer options of yes and no. A correct answer was assigned 1 point, and a wrong answer was assigned 0 points, resulting in a total score ranging from 0 to 7 points. $\alpha$ is a constant, $\beta_1 - \beta_{20}$ is the regression coefficient, $X_i - X_{20}$ represents the explanatory variables (Table 1), and $\varepsilon$ is the error term. Probit and Poisson regression models were tested for multicollinearity and homogeneity by using the Variation Inflation Factor (VIF) and normality of the residuals (Table A1); no signs of homogeneity and multicollinearity were found, as no value exceeded the threshold of VIF > 5, which would be a sign of multicollinearity among the explanatory variables [53]. As the study used a multi-stage sampling procedure, we adjusted the standard error for clustering at the ward level to eliminate the potential unobserved components in outcomes for units within clusters that are correlated [54].

## 4. Results and Discussion

### 4.1. Climate Change Perception in Dry and Humid Zones

Table 3 presents the farmers' climate change perceptions based on indicators of climate change and risk occurrences (from strongly disagree to strongly agree (1–5 point scale). Perceived increases in temperature had a mean of 4.03, indicating that most farmers perceived temperature increases in the last 10 years. These findings agree with [55,56]. Farmers also perceived a decrease in rainfall and a delay in the onset of rainfall. The perception value of farmers in dry AEZs was 3.82, while the mean perception of the humid AEZ farmers was 3.72.

**Table 3.** Climate change perception of indicators and risk occurrences in last 10 years.

| Indicator [1] | Dry AEZs [1] | Humid AEZs [1] | Sig. | Mean and Standard Deviation |
|---|---|---|---|---|
| *Climate change indicator perception* | | | | |
| Increase in temperature | 4.02 (0.98) | 4.04 (0.77) | 0.647 | 4.03 (0.88) |
| Decrease in rainfall (amount) | 3.9 (1.07) | 3.85 (1.00) | 0.241 | 3.77 (1.10) |
| Delay in coming of rainfall | 3.81 (1.22) | 3.72 (1.07) | 0.083 | 3.88 (1.04) |
| *Climate risk occurrence perception* | | | | |
| Increase in frequency of drought | 3.83 (1.07) | 3.88 (0.87) | 0.780 | 3.85 (0.98) |
| Increase in frequency of flooding | 3.84 (0.99) | 3.87 (1.04) | 0.715 | 3.86 (1.01) |
| Increase in evaporation/rapid dry of soil | 3.82 (1.02) | 3.89 (0.84) | 0.857 | 3.86 (0.93) |
| Increase in crop pest and disease outbreak | 4.18 (0.91) | 3.95 (0.84) | 0.000 | 4.07 (0.88) |

Note: [1] measure in five Likert scale from lowest (1 = not perceived) to highest (5 = highly perceived).

Furthermore, farmers perceived an increase in drought, evaporation, and frequency of floods in the last 10 years. These perceptions are in conformity with [17,57]. In addition to climatic conditions, farmers perceived an increase in crop pest and disease outbreaks in the last 10 years. A significant difference between the zones was observed, as 4.18 was the mean perception of farmers of increases in crop pest and disease outbreaks in dry AEZs, while 3.95 was the mean perception of farmers in humid AEZs. Further results revealed no significant differences between the two AEZs in the climate change indicator perceptions, except for the delay in coming rainfall. Within the climate risk occurrence perception, a significant difference was only observed in the increase in crop pest and disease outbreaks. These findings clearly show that the farmers in this study are strongly perceiving negative climate change effects despite the varying climatic conditions in the selected AEZs of Nigeria.

### 4.2. Description of Farmers' Knowledge of Farming Practices Related to Climate Change

Table 4 reports a chi-square test of farmers' knowledge of the causes of climate change, comparing dry and humid AEZs. Farmers in dry AEZs are more aware of deforestation being a cause of climate change than farmers in humid AEZs. In the dry AEZs, 78.70% of farmers knew deforestation could cause climate change, while 52.89% of farmers in humid AEZs were aware of this. Although many of the farmers indicated awareness of the issue, this did not stop them from engaging in deforestation, because they considered it as a drought coping strategy [49,58]. It was found that 72.96% of the farmers in dry AEZs were aware of land clearance by bush burning to cause climate change, as opposed to 47.41% of the farmers in humid AEZs. This is in line with the findings of [49], who reported that farmers had no knowledge of the negative impacts of bush burning. In addition, farmers believe this traditionally used method is the most cost-effective option for land clearance [49].

Simultaneously, 39% of all the respondents were aware that fossil fuel emissions from agricultural machinery could cause climate change. However, there is a significant difference between the farmers of the two groups of AEZs. In dry AEZs, 43.52% of farmers knew fossil fuel emissions could cause climate change, while in humid AEZs, only 24.62%

were aware of this. Farmers thus appear to have relatively low knowledge of this issue. Previous research in Malaysia showed that 85% of the public identified fossil fuel emissions as a major cause of climate change, while the opposite was true for farmers in developed countries, with most farmers knowing about the effect of fossil fuel emissions on global warming [59].

**Table 4.** Farmers' knowledge of farming practices as causing climate change (N = 1080).

| Causes | Item | Dry AEZs (%) N = 540 | Humid AEZs (%) N = 540 | Sig. | Total% |
|---|---|---|---|---|---|
| Deforestation | No | 21.30 | 47.11 | 0.000 | 69.67 |
| | Yes | 78.70 | 52.89 | | |
| Land clearance by bush burning | No | 27.04 | 52.59 | 0.000 | 60.1 |
| | Yes | 72.96 | 47.41 | | |
| Fossil fuel emissions | No | 56.48 | 65.37 | 0.000 | 39.0 |
| | Yes | 43.52 | 24.62 | | |
| Methane from livestock | No | 79.26 | 89.44 | 0.000 | 15.57 |
| | Yes | 20.74 | 10.56 | | |
| Inappropriate manure management | No | 78.15 | 87.04 | 0.000 | 17.41 |
| | Yes | 21.85 | 12.96 | | |
| Excessive use of chemical fertilizer | No | 63.52 | 88.52 | 0.000 | 24.0 |
| | Yes | 36.48 | 11.48 | | |
| Use of chemical plant protection and pesticides | No | 58.34 | 61.67 | 0.264 | 40.0 |
| | Yes | 41.66 | 38.33 | | |

Our results further indicate that farmers have low knowledge of methane emissions from livestock production as contributing to climate change. On average, only 15% of the farmers knew about this, with 20.74% in dry AEZs and 10.56% in humid AEZs. This differs from developed countries such as New Zealand, where many farmers are not only aware of the issue but actively engage in feed management using different types of plants to ensure low amounts of methane produced by their animals [51].

Only 17% of farmers knew that inappropriate manure management could cause climate change because of methane and nitrous oxide emissions. We identified a significant difference between the farmers in different AEZs, with 21.85% of farmers in dry AEZs being aware and only 12.96% for humid AEZs. In addition, 24% of farmers knew about the intensive and indiscriminate use of chemical fertilizers as contributing to climate change. Again, we found a significant difference between the dry and humid AEZs, where 36.48% of the dry AEZ farmers knew that excessive use of chemical fertilizer could cause climate change, while only 11.48% of humid AEZs farmers were aware of this issue. This is in line with the findings of the Environmental Protection Agency [60], in which most respondents were not aware that $N_2O$ is one of the harmful GHGs. We found that 40% of the farmers were aware that the use of chemicals for plant protection and pesticides contributed to climate change, with no significant difference between the two AEZ groupings. In a related study, [61] reported that farmers generally tend to be unaware of the negative effect of agrochemicals on the environment. The result indicated that farmers have very limited knowledge that methane from livestock and inappropriate manure management contribute to climate change, irrespective of their AEZ. Although the respondents in dry AEZs had a lower level of education compared to their counterparts in humid AEZs, the farmers in the dry AEZs had significantly more knowledge of climate change causes in almost all dimensions. This is also in line with previous findings indicating that social status and education might not necessarily lead to more knowledge on a specific subject [62].

### 4.3. Climate Change Knowledge and Its Relation to the Perception of Climate Change

There is a relationship between farmers' knowledge of the causes of climate change and their perceptions of climate indicators (Table 5). The perception of an increase in temperature, decrease in rainfall, delay in coming of the rains, frequency of drought, and increase in the frequency of floods are all positively associated with having higher climate change knowledge scores. Overall, these findings show that perception and knowledge of the causes of climate change are positively correlated with each other. However, it is not clear from this study whether climate change perception pushes the farmers to learn more about climate change causes or whether farmers with more knowledge tend to give more attention to the changes or possibly exaggerate the effects. This is a psychological phenomenon that deserves more attention in further studies.

**Table 5.** Relationship between climate change perception and knowledge of causes (N = 1080).

| Climate Change Perception [1] | | | Climate Change Knowledge Score [2] |
|---|---|---|---|
| Perception Indicators | Mean | Standard Deviation | Correlation Coefficient (*r*) |
| Increase in temperature | 4.03 | 0.88 | 0.651 ** |
| Decrease in rainfall (amount) | 3.79 | 1.02 | 0.820 ** |
| Delay in coming of rainfall | 3.88 | 1.04 | 0.634 ** |
| Increase in frequency of drought | 3.76 | 0.98 | 0.556 ** |
| Increase in frequency of flooding | 3.86 | 1.01 | 0.592 *** |
| Increase in evaporation | 3.87 | 1.03 | 0.140 |
| Increase in crop pest and disease outbreaks | 4.07 | 0.89 | 0.671 *** |

Note: [1] measure in five Likert scale from lowest (1 = not perceived) to highest (5 = highly perceived). [2] Regression coefficient and robust standard error are reported, ** $p < 0.05$, and *** $p < 0.01$.

### 4.4. Factors Influencing Awareness of Climate Change and Knowledge of the Causes of Climate Change

The factors that influence general climate change awareness and the knowledge of agricultural practices contributing to climate change are shown in Table 6. Members of a cooperative are significantly more likely to be aware of climate change ($p < 0.05$) and are more knowledgeable about the causes of climate change as compared to farmers who are not members of such a group (Table 6). Similar observations have been made in other studies [49,63–65]. A higher share of non-agricultural income of a farmer significantly increases the probability of climate change awareness and knowledge of climate change causes ($p < 0.01$). This result is in line with [66].

One reason for this may be that farmers who reduce the future risk by diversifying their household income through non-farm income have more likely to experience climate change events at their farm and are more aware of climate change risks than farmers without non-farm income. Ibrahim et al. [18] reported a significant positive influence of non-agricultural income on both the causes and effects of climate change in southwestern Nigeria.

Farmers who received weather information from government extension agents were more likely to be aware of climate change. The most available and reliable source of agricultural information for farmers in Nigeria is an extension agent under the Agricultural Development Programme (ADP) organization, which has nationwide coverage at every stratum from state to ward levels. While this is in line with some studies [18,46], it contrasts with the findings of others [23,26,67], in which extension contacts affected climate change awareness negatively. We thus see varying effects of extension service provision and how the quality of these programs can have an influence on their effectiveness. Farmers receiving weather information from environmental NGOs are significantly more likely to be aware of climate change and have more knowledge of the causes of climate change. Similar results were reported in Mali and South Africa, where environmental NGOs were identified as the most important source of climate change information among farmers [28,65]. These

findings indicate the need for closer collaboration between the public and private sectors concerning the provision of information on climate change issues.

**Table 6.** Determinants of climate change awareness and knowledge (N = 1080).

| Variable | Probit Regression [1] (Awareness) | Poisson Regression [2] (Knowledge) |
|---|---|---|
| *Socioeconomics* | | |
| Sex | 0.031 (0.038) | 0.055 (0.054) |
| Age | 0.002 (0.016) | 0.003 (0.003) |
| Education | 0.007 (0.002) *** | 0.006 (0.004) |
| Farming experience | 0.003 (0.001) | 0.001 (0.003) |
| Farm size | 0.003 (0.004) | −0.001 (0.007) |
| Agricultural income | −0.001 (0.000) | 0.000 (0.000) |
| Non-agricultural income | 0.005 (0.007) | 0.026 (0.009) *** |
| *Weather information sources* | | |
| Government extension agent | 0.014 (0.032) | 0.227 (0.052) *** |
| Environmental NGOs | −0.068 (0.031) ** | 0.082 (0.052) |
| Farmers' cooperatives | −0.014 (0.032) | .092 (0.045) ** |
| University and research institution | −0.083 (0.058) | −0.019 (0.073) |
| Farmers friends | 0.124 (0.027) *** | 0.232 (0.044) *** |
| *Weather information channels* | | |
| Radio | 0.010 (0.001) *** | 0.009 (0.002) *** |
| Television | −0.008 (0.003) *** | 0.004 (0.004) |
| Newspaper | 0.013 (0.045) | 0.054 (0.068) |
| Internet | 0.017 (0.007) ** | 0.002 (0.005) |
| *Climate risk experience in the last 10 years* | | |
| Flooding | 0.083 (0.034) ** | −0.011 (0.010) |
| Drought | 0.033 (0.032) | 0.034 (0.011) *** |
| Windstorm | 0.002 (0.032) | 0.026 (0.010) *** |
| Dry agro-ecological zones | 0.003 (0.038) ** | 0.231 (0.060) *** |
| *F-value* | *0.000* | *0.000* |
| *Pseudo $R^2$* | *0.1915* | *0.061* |

Notes: [1] Marginal effect and robust standard error are reported. [2] Regression coefficient and robust standard error are reported, * $p < 0.10$, ** $p < 0.05$, and *** $p < 0.01$, VIF = variance inflation factors.

Farmers receiving weather information from farmers' cooperatives were significantly more likely to be aware of climate change and more knowledgeable of the causes of climate change. Other studies, such as from Muench et al. [66], De Sousa et al. [67], and Menike and Arachchi [68], uncovered the positive effects of agricultural cooperatives on information access and awareness of climate change among farmers. Cooperatives serve as a common communication platform to stimulate information exchange among farmers. Therefore, receiving weather information from fellow farmers significantly increased the likelihood of a farmer being aware of climate change. In addition, we observed an increase in knowledge of the causes of climate change due to access to information from other farmers. Farmer-to-farmer interaction was also identified as a highly important source of climate change information in Mali [28] and Nepal [66]. We can thus derive a generally close peer interaction in smallholder farming systems. As local farmer cooperatives encourage peer exchange, farmers in the study area should be motivated to join cooperatives. The importance of cooperatives, informal farmer groups, and peer exchange as information sources among Nigerian farmers is evident. This revelation is particularly important because the dissemination rate in agriculture is comparably low [69].

An increase in receiving weather information via radio significantly increased the likelihood of a farmer's awareness of climate change and knowledge of the causes of climate change. Similar findings were reported in the US and South Africa [65,70]. Using television to access weather information had a significant effect on the likelihood of farmers being aware of climate change. This corroborated the findings of Junsheng et al. [45], who reported the substantial contribution of television to climate change awareness. Mass media, such as television and radio, clearly have a smaller effect on climate change awareness than

the institutional factors reported in this study. Nevertheless, they should not be neglected as information sources, particularly in light of the need for access to weather information in rural areas and in communicating with farmers during emergencies such as pest and disease outbreaks, expected floods, windstorms, or wildfires.

Receiving and searching for weather information primarily from the internet positively influenced the likelihood of farmers being aware of climate change. This effect of internet usage on climate change awareness agrees with the findings of Dorothee et al. [70]. An increase in the number of flood experiences of farmers enhanced their knowledge of the causes of climate change significantly. The experience of droughts made farmers more likely to be aware of climate change while it also increased the farmer's knowledge of the causes of climate change. Experiencing windstorms made farmers significantly more likely to be aware of climate change and increased farmers' knowledge of the causes of climate change.

An interesting revelation of this study was that farmers in the dry AEZs (the Semi-arid, Sudan savannah, and Guinea savannah zones) were more likely to be aware of climate change and had more knowledge on climate change compared to farmers in the humid AEZs (the Rainforest, Mangrove, and Swamp forest zones). This can be attributed to the fact that farmers living in vulnerable climate-risk areas experience the effects of climate change more than those that are not living in climate-risk areas, as depicted by the second argument of the knowledge gap theory [15,33]. Location was found to affect climate change knowledge, including perceived changes in drought, flood, temperature, and rainfall patterns [63]. Similar findings from Kenya and Bangladesh showed that farmers in arid and semi-arid areas perceived a decrease in rainfall and an increase in rainfall variability and temperature more than their humid AEZ counterparts [23,27,41]. This result emphasizes the importance of considering regional differences in the context of climate change awareness campaigns, policy formulation, and mitigation efforts in agriculture. Climate change policies should thus not only be formulated on a national level but specified according to regional requirements.

## 5. Conclusions

This study, drawing on a primary data survey using a structured questionnaire, aimed to (i) assess the knowledge of farmers on farming practices related to climate change and how it is associated with the climate change perception of farmers; (ii) determine the factors influencing awareness and knowledge of climate change. With respect to the causes of climate change attributed to agriculture, we were able to uncover varying degrees of knowledge in our sample. Most respondents knew that deforestation and land clearance by bush burning contributed to climate change. However, many farmers did not know that methane emissions from livestock (enteric fermentation) can cause climate change, despite it being a major GHG contributor within the agricultural sector. This also holds for the inappropriate use of manure, fossil fuel emissions from agricultural machinery, and the excessive and indiscriminate use of agrochemicals.

The climate change knowledge of farmers was found to be positively associated with the climate change perception of farmers. This finding serves as evidence that wrong or missing information can lead to distorted perceptions. Critical gaps in knowledge consequently lower the mitigation preparedness of farmers towards climate change. Given the mixed results in the level of knowledge about the agricultural causes of climate change among the respondents, we recommend policymakers to focus on educating farmers about the effects of farm practices on the environment. A well-planned process of knowledge transfer would positively influence the degree of understanding of the subject matter.

Regarding the factors positively influencing the awareness of climate change and knowledge of its causes, contrary to the first aspect of the knowledge-gap theory, socioeconomic factors did not have a great effect on farmers' climate change awareness and knowledge of farm practices that mitigate climate change. This may be because the smallholder farmers seem to be socioeconomically homogenous. However, weather information

sources, channels, and climate risk experience of farmers had a significant influence on the farmers' climate change awareness and knowledge of farm practices mitigating climate change. Cooperative membership, government extension agents, environmental NGOs, and farmer-to-farmer climate change information sources influenced the farmers' climate change awareness and knowledge of farming practices that mitigate climate change.

This indicates the importance of using subject information sources in teaching farmers the effects that farming practices, such as methane from livestock and inappropriate manure management, can have on the climate. As radio affects both the farmers' climate change awareness and knowledge of farming practices that mitigate climate change, it is important for creating climate change awareness and guiding climate change knowledge of farmers toward scientific facts and findings for appropriate action. In 2022, the Nationally Determined Contributions (NDC) of Nigeria reported that land-use emissions, together with agriculture, represent about one-quarter of emissions [71]. Thus, this sector holds significant climate change mitigation potential for Nigeria through reductions of GHGs as well as the enhancement of agricultural sequestration and application of climate-smart technologies. The government and other actors, such as Climate Change Network Nigeria (CCN-Nigeria), and funding bodies such as the Green Climate Fund and Adaptation Fund should be tasked with spreading knowledge among farmers and supporting appropriate land use practices (such as agroforestry), which will increase the carbon sink potentials of the sector.

Furthermore, as experiencing drought, floods, and windstorms were identified as positive drivers of climate change awareness and knowledge, we found farmers of humid AEZs were less knowledgeable about the farm practices that mitigate climate change than their peers in dry AEZs. Living in areas prone to a higher climate risk thus also increases the level of climate change knowledge. This is particularly true when there is no large difference in income or education as well as access to information sources and channels among the respondents, as depicted by the knowledge gap theory. Therefore, we identified the location as an important factor in framing the perception and knowledge of climate change. These findings indicate that farmers of climate risk-prone areas are ahead of their counterparts in terms of climate change perception and knowledge of farming practices that mitigate climate change, which will ease the adaptation process of being guided toward specific practices. This highlights the need to give a special focus to agricultural impacts in the savanna zones and semi-arid regions of the country, which are most affected by the impacts of climate change.

Climate change awareness and education schemes should be made available through farmers' cooperatives, radio, television, and the internet. The better the farmers understand climate change issues and how they affect them, the more they will be ready to adapt to them accordingly. An increase in organizational involvement with farm-related associations and encouragement of farmers to participate in farmer-to-farmer extension and knowledge-sharing networks could strengthen their climate change knowledge and shape their perceptions. This study does not investigate the effect of sociocultural factors such as religion, which may influence the farmers' perception [72], as the country is one of the countries where religion is considered very important in people's daily life [73]. We recommended that sociocultural factors should be considered in a future climate change study.

**Author Contributions:** M.Y.M. conceptualization, data collection, analysis, writing of the original draft; S.M. visualization and literature review; H.K. validation and visualization; M.B. supervision, funding acquisition and project administration. All authors have read and agreed to the published version of the manuscript.

**Funding:** This research was funded by Internally Grant Agency of Faculty of Tropical AgriScience, (Grant number; 20213102) Czech University of Life Science Prague, Czech republic and the APC is paid by Leibniz Centre for Agricultural Landscape Research (ZALF), Eberswalder Straße 84, 15374 Müncheberg, Germany; harald.kaechele@zalf.de.

**Conflicts of Interest:** The authors declare no conflict of interest.

## Appendix A

**Table A1.** Variance inflation factor test of multicollinearity.

| Variable | Variance Inflation Factor Coefficient | 1/Variance Inflation Factor |
|---|---|---|
| *Socioeconomics* | | |
| Sex | 1.15 | 0.867 |
| Age | 2.94 | 0.340 |
| Education | 1.46 | 0.686 |
| Farming experience | 2.55 | 0.392 |
| Farm size | 1.23 | 0.813 |
| Agricultural income | 1.05 | 0.953 |
| Non-agricultural income | 1.16 | 0.864 |
| *Weather information sources* | | |
| Government extension agent | 1.26 | 0.794 |
| Environmental NGOs | 1.20 | 0.834 |
| Farmers' cooperatives | 1.25 | 0.799 |
| University and research institution | 1.21 | 0.824 |
| Farmers friends | 1.22 | 0.820 |
| *Weather information channels* | | |
| Radio | 1.36 | 0.736 |
| Television | 1.22 | 0.823 |
| Newspaper | 1.15 | 0.867 |
| Internet | 1.22 | 0.817 |
| *Climate risk experience in the last 10 years* | | |
| Flooding | 1.25 | 0.801 |
| Drought | 1.37 | 0.727 |
| Windstorm | 1.24 | 0.804 |
| Dry agro-ecological zones | 2.69 | 0.371 |

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
