# Peer review of "Climate Change Knowledge and Perception among Farming Households in Nigeria"

_climate, doi:10.3390/cli11060115_

Round 1
Reviewer 1 Report
GENERAL: The subject of this manuscript is intensively researched, also in Africa. Nevertheless the manuscript shows some interesting, partially original results.
Over-all, the text is well cared presented, written and accessible.
INTRODUCTION: Good ands relevant; rather long there are oportunities to shorten and to bring the textr sharper and closer to the subject and the aims of the text.
MATERIALS AND METHODS: The used approach is not conventional. Why were these M&M selected? Provide examples of similar studies.
Study area: What are environmental (apart from drought and floods), social, eonomic factors in Nigeria, that may ('theoretically) affect the perception of the farmers? Further on in the text production and windstorms are mentioned.
"Table 1" line 163, does not correspond with "Table 1 line 223
Are all parameters listed in Table 1 on page 6 distributed in a Gaussean way? What is the meaning of the standard deviation? How did you use it?
DISCUSSION: A real discussion lacks. Elements of a discussion as the comparision of some findings with the existing literature are provided. Other essential elements as comments on the strengths and weaknessess of the approach, lack.
MINOR Comments:
-P8 line 267: unnecessary repetition.
- P7-8: Do not break Table 8
-
Author Response
|
Comment |
Response |
|
|
1 |
INTRODUCTION: Good and relevant; rather long there are opportunities to shorten and to bring the text sharper and closer to the subject and the aims of the text. |
Thank you for this suggestion. We shortened the introduction by deleting some peripheral sentences. See line L41-121. |
|
2 |
MATERIALS AND METHODS: The used approach is not conventional. Why were these M&M selected? Provide examples of similar studies. |
We mentioned the reason why we applied a particular sampling method at every stage of our sampling. See L149-160. |
|
3 |
Study area: What are environmental (apart from drought and floods), social, economic factors in Nigeria, that may ('theoretically) affect the perception of the farmers? Further on in the text production and windstorms are mentioned. |
For example, also beliefs may influence farmers' perceptions of climate change, as postulated by the cultural theory of risk for climate change adaptation, which states that “an individual’s perception of their own climate risk is informed by their social interactions and their fundamental beliefs about society and nature” (McNeeley and Lazrus, 2014). As Nigeria is one of the top countries where religion is considered very important in people’s daily life (Pew Research Center, 2018) we recommend this aspect to be investigated in a future studies. See L532-536. References McNeeley, S. M. and Lazrus, H. (2014). “The cultural theory of risk for climate change adaptation.” Weather, Climate, and Society. Pew Research Center (2018). The age gap in religion around the world. https://www.pewresearch.org/religion/2018/06/13/how-religious-commitment-varies-by-country-among-people-of-all-ages/ |
|
4 |
"Table 1" line 163, does not correspond with "Table 1 line 223 |
We reconciled it. Thank you for this observation |
|
5 |
Are all parameters listed in Table 1 on page 6 distributed in a Gaussean way? What is the meaning of the standard deviation? How did you use it? |
The continuous variables in the table: farm size, agric. and non-agric. income, radio, and drought experience were found to be in Gaussean/normally distributed.
The standard deviation is expressed by how much our observations are distributed around the mean value. It helps in avoiding models that consider the normal distribution of data as a cardinal assumption. |
|
6 |
DISCUSSION: A real discussion lacks. Elements of a discussion as the comparision of some findings with the existing literature are provided. Other essential elements as comments on the strengths and weaknessess of the approach, lack. |
We included the strengths and weaknesses of our approach. For the strength, see L445-448. For the weakness, see L532-536. |
|
7 |
P8 line 267: unnecessary repetition. |
We deleted the repeated sentence. See L283-284 |
|
8 |
P7-8: Do not break Table 8 |
We rectified them thank you (we hope after the acceptance of the changes, no table will be broken on two pages) |
Reviewer 2 Report
The issues on relationships between of climate change knowledge and perception among farming households in Nigeria are discussed in the paper. Overall, the paper is good, but I suggest the authors to revised as follows:
First, some novel topics should be presented. The current topic is some confusing.
Second, some frameworks should be developed based on some theories, with some hypotheses proposed.
Author Response
|
|
Comment |
Response |
|
1 |
First, some novel topics should be presented. The current topic is some confusion. |
Dear reviewer, thank you for the comment. We selected the topic as we didn’t find comprehensive studies that investigate the issue of climate change knowledge and perceptions of smallholder farmers across all agroecological zones in Nigeria. |
|
2 |
Second, some frameworks should be developed based on some theories, with some hypotheses proposed. |
We developed a conceptual framework for our model that is presented in Figure 1, page 4. In this study, we work with the research question presented at the end of the introduction instead of the hypothesis as usual in studies testing a large number of independent variables to avoid working with a high number of hypotheses. |
Reviewer 3 Report
Dear Authors, please find attached my comments on your manuscript, which needs just minor adjustments to be published.
The manuscript "Climate Change Knowledge and Perception among Farming Households in Nigeria" addresses an important issue, namely the attitude of smallholder farmers toward climate change, their perceptions of the impacts, and their knowledge of the causes related to farming. Especially since the study is conducted in the most populous country in Africa.
The abstract is balanced and sufficiently concise. The introduction provides an overview of the issue and the references needed to frame the study. The methods present the study area, how the sample was identified, the mode of analysis and descriptive statistics. The results, which also include discussion, present the main findings on climate change perception in dry and humid zones, farmers' knowledge of farming practices related to climate change and the factors influencing awareness of climate change and knowledge. Methods and results are not particularly innovative, as demonstrated by the convergence with previous studies, however contribute to clarify how knowledge and awareness on climate change is penetrating the rural world in developing countries. This, rather than through new technologies comes through peer-to-peer relationships, thus farmer to farmer, which turns out to be one of the most powerful means of creating awareness and educating farmers. The conclusions are a bit long however they make the point and include appropriate recommendations.
The paper is well written, fluent and grammatically correct.
To publish the paper just minor revision are needed:
Line 12: land use: land use itself is not a cause of emissions increase, rather than land clearance, land cover changes and so on. Please change
Line 67: the same than above
Line 87: the sentence seems out of context
Line 253: please move the table 2 after the last bullet
Line 544: the link is not working
Line 554: please add the doi
Line 598: the link is not working
Few other punctuation typing errors should be checked before publishing. Check all the links of references.
See also the annotated pdf I attach

Author Response
|
Comment |
Response |
|
|
1 |
Line 12: land use itself is not a cause of emissions increase, rather than land clearance, land cover changes and so on. Please change |
We changed this. Thank you for the correction. |
|
2 |
Line 67: the same than above |
We corrected it. See L68 |
|
3 |
Line 87: the sentence seems out of context |
Is deleted. See L94 |
|
4 |
Line 253: please move the table 2 after the last bullet |
We moved the table to the suggested position. See L245-246 |
|
4 |
Line 544: the link is not working |
We provided the workable link as: https://www.climatelinks.org/resources/greenhouse-gas-emissions-factsheet-nigeria. See L565 |
|
5 |
Line 554: please add the doi |
We provided the doi number as; https://doi.org/10.1108/IJSE-09-2013-0201. See L575 |
|
6 |
Line 598: the link is not working |
We provided the working link as : https://www.un.org/africarenewal/magazine/november-2021/dogged-massive-floods-nigeria-ramps-actions-tackle-climate-crisis#:~:text=The%20wall%2C%20spanning%201%2C500%20kilometers,%2C%20Jigawa%2C%20Yobe%20and%20Borno See L619-621 |
|
7 |
Few other punctuation typing errors should be checked before publishing. Check all the links of references. |
We rechecked the manuscript once again, and all typing errors and reference links were rectified appropriately. Thank you for the time given to go through our paper and for giving vital suggestions on how to improve our work. |